# Pathogenicity of PKCγ Genetic Variants—Possible Function as a Non-Invasive Diagnostic Biomarker in Ovarian Cancer

**DOI:** 10.3390/genes14010236

**Published:** 2023-01-16

**Authors:** Kanza Shahid, Khushbukhat Khan, Yasmin Badshah, Naeem Mahmood Ashraf, Arslan Hamid, Janeen H. Trembley, Maria Shabbir, Tayyaba Afsar, Ali Almajwal, Ali Abusharha, Suhail Razak

**Affiliations:** 1Department of Healthcare Biotechnology, Atta-Ur-Rahman School of Applied Biosciences, National University of Sciences and Technology, Islamabad 44010, Pakistan; 2School of Biochemistry and Biotechnology, University of the Punjab, Lahore 54590, Pakistan; 3LIMES Institute (AG-Netea), University of Bonn, Carl-Troll-Str. 31, 53115 Bonn, Germany; 4Minneapolis VA Health Care System Research Service, Minneapolis, MN 55417, USA; 5Department of Laboratory Medicine and Pathology, University of Minnesota, Minneapolis, MN 55455, USA; 6Masonic Cancer Center, University of Minnesota, Minneapolis, MN 55455, USA; 7Department of Community Health Sciences, College of Applied Medical Sciences, King Saud University, Riyadh 11362, Saudi Arabia; 8Department of Optometry, College of Applied Medical Sciences, King Saud University, Riyadh 11362, Saudi Arabia

**Keywords:** protein kinase c-gamma (PKCγ), ovarian cancer, protein-protein docking, molecular dynamics simulation

## Abstract

Ovarian cancer has the highest mortality rate among gynecologic malignancies, owing to its misdiagnosis or late diagnosis. Identification of its genetic determinants could improve disease outcomes. Conventional Protein Kinase C-γ (PKCγ) dysregulation is reported in several cancers. Similarly, its variant rs1331262028 is also reported to have an association with hepatocellular carcinoma. Therefore, the aim of the present study was to analyze the variant rs1331262028 association with ovarian cancer and to determine its impact on PKCγ’s protein interactions. Association of variation was determined through genotyping PCR (cohort size:100). Protein–protein docking and molecular dynamic simulation were carried out to study the variant impact of PKCγ interactions. The study outcome indicated the positive association of variant rs1331262028 with ovarian cancer and its clinicopathological features. Molecular dynamics simulation depicted the potential influence of variation on PKCγ molecular signaling. Hence, this study provided the foundations for assessing variant rs1331262028 as a potential prognostic marker for ovarian cancer. Through further validation, it can be applied at the clinical level.

## 1. Introduction

In women diagnosed with gynecological cancers, ovarian cancer is the primary cause of death. Worldwide, it is the fifth most common cause of death in women. The current screening tests have a low predictive value, and the majority of cases are discovered at an advanced stage, resulting in poor disease outcomes [1,2,3]. According to WHO statistics for 2020, around 313,959 cases of ovarian cancer were detected worldwide with a total of 207,252 deaths, which makes up about 3.4% of total cancer cases reported. The 5-year prevalence rate of ovarian cancer was also high with about 823,315 cases reported worldwide. In Pakistan, around 90,000 cases of ovarian cancer were reported in 2020 [4]. Risk factors for ovarian cancer include smoking, menopause, hormone replacement therapy (HRT), endometriosis, and *BRCA1* and *2* mutations [5].

PKCγ belongs to the large family of PKC serine/threonine-specific protein kinases which are involved in various cellular and signal transduction pathways [6,7]. Calcium-dependent PKCγ is encoded by the *PRKCG* gene located on chromosome 19 at position 19q13.2-q13.4 [8]. Along with PKCα and PKCβ, *PKCγ* is a conventional PKC (cPKC) isoform involved in cancer progression. PKCα was found to promote colon cancer [9] and breast and ovarian cancer [10], and PKCβ was implicated in the progression of breast cancer [11], colon cancer [12], prostate cancer [13], and glioblastoma [14]. PKCγ was shown to play roles in colon cancer [15], osteosarcoma [16], and glioblastoma [17].

Known as a single nucleotide polymorphism (SNP), the substitution of one nucleotide for another is the most basic form of DNA variation among individuals. SNPs in the gene’s coding region often have a deleterious effect that can lead to cancer development [18,19,20]. Missense SNPs can cause cancer as they affect the protein’s capacity to bind its substrate or inhibitors as well as its subcellular location [21]. There are many SNPs identified in the genes *BRCA1*, *BRCA2*, *AURKA*, *CYP19A1*, *SRD5A2*, *RB1*, *CDK6*, *CDK12*, *RAD51*, *XRCC2*, *NM1*, *CCND1* and *TP53* that are associated with ovarian cancer [22,23,24]. Genetic polymorphisms in PKCγ affect the protein structure and function, thus potentially contributing to progression toward cancer [25]. Several SNPs in different PKC isoforms (rs454006, rs2242245, and rs8103851 of PRKCG, rs11079651 in PRKCA, and rs34367566 in PRKCB) have been reported to be linked with various types of cancer [16,26]. Recent reports showed that missense variants in a PKCγ family member, PKC-epsilon (PKCε), are associated with altered protein structure and function as determined using different databases [27]. Similarly, a study recently indicated the *PRKCG* variant rs1331232028 association with HCV-induced hepatocellular carcinoma [28]. The deleteriousness of the variant rs1331232028 was predicted through multiple consensus tools that along with variant rs1331232028 also predicted the deleteriousness of variant rs923331350 [28].

Therefore, in this study, the non-synonymous SNPs (nsSNPs) rs923331350 and rs1331232028 in *PRKCG* that result in amino acid replacement in the zinc ion binding domain (A24S) and ATP binding domain of the protein (K359R) were selected and genotyped in patient and control groups to investigate the possible association with ovarian cancer in the Pakistani population. This study will build a foundation for delineating PKCγ as a prognostic marker in ovarian cancer that will facilitate in designing a non-invasive diagnostic method for ovarian cancer.

## 2. Materials and Methods

### 2.1. Collection and Processing of Samples

Institutional Review Board approval (IRB no = 10-2021-01/01, provided as Appendix A) was acquired from the parent department ASAB of the National University of Sciences and Technology (NUST) before the start of the current investigation. Before collecting the blood samples, the patients gave their written and verbal consent (given as Appendix A). Blood samples from a total of 49 patients diagnosed with ovarian cancer were collected from CMH hospital, Rawalpindi. Blood samples from 51 healthy people were also collected. Approximately 3–5 mL of blood was extracted using sterile syringes into 5 mL ethylene diamine tetra acetic acid (EDTA) tubes.

The inclusion and exclusion criteria were set for the study. The current study only included patients who had been clinically diagnosed with localized and/or metastatic ovarian cancer and were currently receiving chemotherapy or radiotherapy treatment. Patients whose tumors were cleared and who just visited the hospital for follow-up were excluded from the study. Furthermore, patients who had another metabolic, cardiac, or neurological disease were also excluded from the study.

### 2.2. Genomic DNA Extraction and Genotype Analysis

DNA was isolated from entire blood samples of research participants using an organic (phenol–chloroform) extraction procedure [29]. Primer 1 [30], a bioinformatics tool, was used to create the primers for SNP analysis. Two outside primers and two interior primers were constructed for the gene in such a way that they must amplify the specific area in the gene bearing the targeted variant. UCSC in silico PCR [31] was also used to validate the primers. Tetra-ARMS PCR was used to genotype the nsSNPs rs923331350 and rs1331232028 in Veriti™ 96-Well Thermal Cycler of Applied Biosystems™. Solis BioDyne FIREPol^®^ Master Mix with 7.5 mM MgCl_2_ was used to make a reaction mixture of 25 µL for each sample. The primer sequences and conditions for PCR reaction are given in Table 1. Agarose gel electrophoresis was used to examine the amplified product. The gel used was 2% *w*/*v* for the PCR product and visualized under a UV transilluminator.

### 2.3. In-Situ Mutagenesis

The 3D structure of PKCγ is predicted through I-TASSER [32]. In situ mutagenesis was carried out by replacing the wild type of amino acid in the original PKCγ structure predicted by I-TASSER with the variant amino acid. PyMol, a molecular visualization system, was used to introduce the A24S (rs923331350) and K359R (rs1331232028) amino acid variation [33].

### 2.4. Statistical Examination

GraphPad Prism software 9 [34] was used to perform statistical analysis on the genotyping data obtained. On both the ovarian cancer patients and control samples, the Chi-square test was used. The analysis of odds ratios and relative risk was also carried out with their corresponding confidence intervals defined. A *p*-value of less than 0.005 was considered statistically significant. The effect of SNPs on protein structure and function was also checked by seeing how the SNP affect the mRNA structure through RNAfold Web Server [35].

### 2.5. Molecular Docking of PKCγ with Connexin43

To determine the impact of amino acid variants in PKCγ on its interactions with target proteins, molecular docking was performed. The 3D structure of Connexin43 was predicted from I-TASSER [32] and was used for molecular docking simulations. Molecular docking of PKCG with one of its binding partners, Connexin 43 or GJA1, was performed using HADDOCK 2.4 [36] to assess the molecular effect of missense SNP on protein–protein interactions of PKCγ. This protein against PKCγ was chosen following a thorough analysis of the literature and the use of protein–protein interaction databases such as STRING [37]. The HADDOCK structures were visualized using Ligplot+ [38].

### 2.6. Interaction Dynamics Analysis

After the docking performed through HADDOCK, the dynamics of molecular interactions between PKCγ variants with Connexin43 were investigated using GROMACS [39] and the OPLS-AA force field [40], which was used to simulate wild-type and variant PKCγ/Connexin43 complexes. A cubic box was formed around each complex for solvation by incorporating SPC216 water molecules, which was followed by neutralization by incorporating Na^+^/Cl^−^ ions. MD simulations were initially energy minimized for 50,000 steps, which was followed by NVT and NPT equilibration for 100 ps. MD complex trajectories were started with the same random seed. MD simulations of wild-type and variant complexes were carried out for a 10 ns production run. GROMACS 2016 in-built programs (gmx_trjconv) were used to create dynamic trajectories. The bond length analysis among amino acids of the protein–protein interaction was completed using VMD [41]. A variety of complex structural analyses was also carried out. The gmx_rms command was used to calculate the root mean square deviation (RMSD for protein backbone), the gmx_ rmsf command was used to calculate the root mean square fluctuations (RMSF for protein, side chains), the gmx_ gyrate command was used to calculate the radius of gyration (Rg for protein, backbone), and the gmx_ sasa and gmx_ hbond commands were used for surface area (SASA) number of hydrogen bonds analysis. The scatter line plot was used to represent all of the MD analysis.

### 2.7. Pathway Construction for PKCγ’ and Connexin43 Interaction

The interactive pathway is built to examine the effect of wild and mutant proteins on PKCγ’s interaction with Connexin 43. The interactive pathway is built to examine the effect of wild and mutant proteins on PKCG’s interaction with Connexin 43. The interactive pathway of PKCγ and Connexin 43 was created after reviewing the literature and various databases such as KEGG [42] and STRING [37].

## 3. Results

### 3.1. Clinico-Pathological Characteristics of Ovarian Cancer Patients

Characteristic information on the ovarian cancer patients included in our analysis was collected to take into consideration important aspects such as age, cancer stage, metastasis state, and treatment stage. The clinicopathological characteristics of the ovarian cancer patients are shown in Table 2 and Appendix A.

### 3.2. Association of A24S (rs923331350) and K359R (rs1331232028) SNPs of PRKCG with Ovarian Cancer

The DNA extracted from ovarian cancer (*n* = 49) and control (*n* = 51) samples was genotyped for the presence of missense variants rs923331350 G/T and rs1331232028 G/A in the PRKCG gene; these SNPs result in the substitution of Alanine (A) to Serine (S) at residue 24 and Lysine (K) to Arginine (R) at position 359, respectively. Tetra ARMS-PCR was used for the genotyping. This method employs four primers to amplify the targeted gene sequence, resulting in an outer control band and an inner genotype band. Table 3 shows the band sizes for both nucleotide variants.

The frequency distribution for genotypes of both PKCγ genetic variants rs923331350 and rs1331232028 for control and ovarian cancer samples was calculated. It was found that for variant rs923331350, only the heterozygous AG allele was found in both control and patient samples, suggesting that the results are null in wet lab experimentation. No further analysis of A24S SNP was carried out. The results of genotyping data of A24S are shown in Appendix A.

For variant rs1331232028, genotype AA was found to be statistically significant with an odds ratio of 2.508 and relative risk of 1.535, along with a *p* value of 0.0515, and it was associated with increased risk of ovarian cancer, whereas genotype GG was also statistically significant, but instead of increased risk, it was displaying a protective effect against ovarian cancer, as shown in Table 4.

The statistical analysis revealed that the frequency of the A-allele in variant rs1331232028 was significantly higher in the ovarian cancer group compared to the control group. However, the G-allele showed the opposite pattern and was found to be significant and statistically higher in the control group compared to the ovarian cancer group (Table 4).

### 3.3. Association of K359R (rs1331232028) SNP of PRKCG with Metastatic State and Stage of Ovarian Cancer

The genotypic distribution of K359R SNP with different clinical features of ovarian cancer was investigated to further evaluate the association of K359R SNP with ovarian cancer (Table 5). The wild-type allele AA is associated with metastatic cancer with an odds ratio of 0.41 and relative risk of 0.52 (*p* = 0.0017), while genotypes AG and GG show no association with metastatic ovarian cancer. There is no associated allele found with non-metastatic ovarian cancer.

When considering the genotype distribution in different stages of ovarian cancer, the wild-type allele AA demonstrates an odds ratio value of 5.19 and relative risk of 3.19 in association with stages I–II of ovarian cancer (*p* = 0.0065), while the mutant allele GG with less than 1 value of the odds ratio of 0.20 and relative risk of 0.28 shows a protective role against ovarian cancer stages I–II (*p* = 0.02). None of the alleles showed an association in stages III–IV of ovarian cancer.

### 3.4. Influence of (SNP rsIDs rs923331350 and rs1331232028) on the PRKCG mRNA Secondary Structure

In silico analysis was performed to predict the mRNA secondary structure for wild-type and variant alleles of PKC. The Minimum Free Energy (MFE) values for both wild-type and mutant variants were analyzed. The secondary structure of mRNA for rs923331350 G/U and rs1331232028 A/G showed dramatic changes in the structures when compared to wild type, demonstrating the significant effect of a variant allele on the overall structure of mRNA (Figure 1A,B). For the variant rs923331350, the MFE value for the reference allele G and variant allele T was −7.4 Kcal/mol. In the case of rs1331232028, the wild-type allele A showed an increase in MFE value (−1.40 kcal/mol) as compared to the variant allele G with an MFE value of −3.80 Kcal/mol. In the case of variant rs923331350, both the wild-type and mutant alleles have the same stability due to the same MFE value, whereas in variant rs1331232028, the wild-type allele is less stable than the mutant allele due to its increased MFE value. The wild-type allele A in variant rs1331232028 is associated with ovarian cancer risk (Figure 1C,D), which might be due to the result of decreased mRNA structure stability.

### 3.5. Influence of PRKCG SNPs on PKCγ–Connexin 43 Interaction

Among the clusters obtained after performing HADDOCK docking analysis, the top cluster was selected, and within that cluster, the structure with a low z-score was selected and the results were visualized using LigPlot+. The docking parameters of wild-type and mutant proteins upon which the top cluster are selected for analysis (Appendix A). The docking parameters include different factors such as Van der Waals energy, electrostatic energy, desolvation energy, restraint violation energy, and buried surface area.

Wild-type and both mutant proteins were docked with connexin 43 (also known as GJA1), and their interaction with connexin 43 is shown in Figure 2A. There were 13 hydrogen bonds between wild-type protein and connexin 43. There were hydrophobic interactions between wild-type and connexin 43, as shown in Figure 2a below. In wild-type interaction with connexin 43, Lys503 is bonded with Val 539 of connexin 43 and ARG 513 is bonded with SER 325 of connexin 43.

The hydrogen bonds in the mutant proteins are changed as compared to wild type. The position of bonds is slightly changed, and the number of bonds is increased in mutants as well. There were 13 hydrogen bonds between wild-type PRKCG and connexin 43, whereas there were 15 hydrogen bonds between A24S mutant and connexin 43 (Figure 2B). The hydrophobic interactions are more with mutant A24S as compared to wild type. In mutant A24S protein, ARG 374 is bonded to ASP 666, whereas in wild-type protein, ARG 374 is bonded with ASN 473 and GLU 603. Likewise, LYN 346 is bonded with ASP 480 in mutant A24S protein but in wild type, it is bonded with TYR 529.

The different bonding pattern is also seen in wild-type and K359R mutant protein (Figure 2C). There were 14 hydrogen bonds in the K359R mutant protein. In the wild-type interaction, ASN 473 is bonded with ALA 371 of connexin 43, whereas in K359R mutant, ASN 473 is bonded with PRO 355 of connexin 43. There was previously no salt bridge between wild type and connexin 43, but one was formed between K359R mutant residue ARG 615 and connexin 43 residue ASP 378. The formation of a salt bridge indicates the stronger bonding of the K359R mutant with connexin 43. Overall, the mutant K359R–connexin 43 interaction was the more stable, allowing for the maximum protein–protein interaction.

### 3.6. Interactions Dynamic Analysis of Wild-Type and Variant PKCγ–Connexin 43 Complexes

For the detailed analysis of the protein–protein interaction among wild-type and mutated proteins, MD analysis was conducted. Despite there being no association of variant rs923331350 with disease, we have performed a further analysis on this variant to gain potential insight behind its impact on the protein–protein interaction. Following the simulation, several files containing critical data were generated for both wild-type and mutated protein–protein interactions. The data in those files were plotted on graphs to help understand the simulation results of the interaction among proteins and the effect mutation has had on the protein–protein interaction. The following characteristics were used to examine the differences between wild-type and mutant proteins: the analysis of the root mean square deviation (RMSD), root mean square fluctuation (RMSF), radius of gyration, number of hydrogen bonds, and solvent accessible surface area (SASA). Their graphs are plotted in scatter plot.

RMSD analysis of the interaction of the wild-type and mutant protein with connexin 43 revealed that the altered protein deviates when compared to the wild type (Figure 3A). The lower the RMSD value, the more stable the interaction. The RMSD value from 0 to 4 ns was 0.6 nm for all three interactions. After 4 ns, the RMSD value of the three interactions started to deviate from 0.6 nm, and at the end of the simulation, i.e., 10 ns, the wild-type interaction has a 0.8 nm value, A24S has a 0.6 nm value while K359R has a 0.5 nm value. The wild-type protein interaction with connexin 43 has a higher RMSD value than the mutated protein, showing that the interaction is stronger with the mutated protein. Among the mutated proteins, the overall interaction is quite similar as shown in the peaks for both proteins. So, the mutated protein interaction with connexin 43 was stronger and more stable.

At the start of the simulation, the wild-type interaction had a 600 nm^2^ SASA value, while both mutants have a 580 nm^2^ SASA value. The SASA value for wild-type interaction increases and reaches 630 nm^2^ at 6 ns, and after that, the value decreases to 590 nm^2^ at 10 ns. The A24S interaction SASA value started to decrease (560 nm^2^) after 1 ns, and at the end of the simulation, the value did not change much (i.e., 580 nm^2^). In the case of K359R interaction, the SASA value increased after 1 ns going to 610 nm^2^; after that, the value decreased until 7 ns at, which point the value increased again to 610 nm^2^. After that interval, the SASA value of K359R interactions decreased, ending at 580 nm^2^. Both mutant protein interactions have lower SASA values than the wild-type interaction at the beginning of the stimulation, and the trend remains the same toward the end of the stimulation. The data indicate that wild-type PKCγ is loosely bound with connexin 43, whereas the mutant A24S and K359R proteins bind rigidly with connexin 43 throughout the stimulation (Figure 3B).

The radius of gyration, abbreviated as Rg, was calculated for wild-type and modified protein interactions (Figure 3C). The wild-type protein interaction with connexin 43 started at 4.3 nm, and throughout the simulation, it reaches 4.25 nm and 4.4 nm, ending at 4.38 nm. The wild-type protein interaction has a larger overall gyration radius than the A24S and K359R mutants, which shows its loose interaction with connexin 43. The radius of the A24S mutant interaction gyration value decreases after 10 ns of stimulation, starting from above 4.1 nm and ending at 4.0 nm. The K359R mutated interaction after rise and fall during the 10 ns stimulation gives an almost similar gyration value. Both proteins have more stable interaction with connexin 43.

Hydrogen bonds play a critical role in the molecular interactions and structures of proteins. The number of hydrogen bonds that are formed in protein structures during simulations were determined to analyze the effect of the variants on protein interactions. During the period of 0–2 ns, the number of hydrogen bonds decreased from 700 to 600, but after that, the interval number of hydrogen bonds remained stable throughout the simulations. It can be seen in Figure 3D that the number of hydrogen bonds in mutant complexes is slightly higher compared to that in the wild-type complex, showing that the molecular interactions that are formed between A24S and K359R are more stable as compared to the wild type.

The bond length among different interactive amino acids of both wild-type and mutant PKCγ with connexin 43 was examined to see whether the bond length changes over 10 ns stimulation or not. Three pairs of such interacting residues were randomly selected, and the pictures were taken using different intervals for bond length such as 0 ns, 5 ns and 10 ns for all three interactions, and are shown in Table 6. In the wild-type PKCγ–connexin 43 complex, the distance between the interacting residues Gln48–Arg374 and Tyr529–Asn341 was increased from 0 to 10 ns; however, the distance between Lys503–Val359 was decreased at the end of the 10 ns simulation, showing the interactive amino acids came closer to each other (Figure 3E). Distances in A24S–connexin 43 complex interacting residues were determined. The distance in two interactive pairs Ser639–Pro334 and Cys516–Glu352 was decreased, while the distance between one interactive pair Ser664–Arg376 remains the same at the time interval of 0 ns and 10 ns, showing (Figure 3F) that closer and more stable bonds are formed. Similarly, for the K359R–connexin 43 complex, the distance between only one pair of interacting residues Phe469–Arg362 has decreased, whilst the distance between the other two pairs Arg413–Ser 328 and Arg615–Arg376 increased (Figure 3G).

### 3.7. PKCγ and Connexin 43 Interaction Pathway

The PKCγ interacts with connexin 43 in various types of cells in the human body, resulting in the regulation of gap junctions in cells by allowing small molecules to pass through. The wild-type PKCγ phosphorylates the gap junction and causes it to be down-regulated [43].

The signaling molecule makes entry into the cell via different receptors such as G-coupled protein receptors (GPCRs) and receptor tyrosine kinase (RTK) in the ovaries. Histamine, chemokine, and luteinizing hormone are ligands for various GPCRs such as the histamine 1 receptor, CXCR4 receptor, and luteinizing hormone receptor, which enter the cell and activate phospholipase C (PLC) [44]. Similarly, PLC is activated by various RTK receptors such as EGFR, VEGFR, FGR and IR upon agonist binding [45]. PLC binds to phosphatidylinositol 4, 5-bisphosphate (PIP2) after activation, producing IP3 and DAG. IP3 binds to the IP3R receptor in the endoplasmic reticulum, causing calcium production to increase. Both DAG and Ca^2+^ cause PKCɣ activation. The phosphorylation of specific Ser or Tyr residues in the C-terminal domain of connexin 43 has been shown to reduce gap-junctional intercellular communication, and PKCɣ phosphorylates connexin 43 mostly at Ser-368 and Ser-372 in the C-terminal domain. The phosphorylation of Ser-368 by PKC results in a conformational change in the C-terminal domain, which reduces the permeability of connexin 43 GJH to small organic solutes [46,47].

In the case of A24S and K359R mutant PKCɣ, they phosphorylate connexin 43 more rapidly, resulting in a complete blocking of gap junctions (Figure 4). The gap junction’s blockages cause no passage of ions, solutes and small molecules among cells, leading toward a loss of cell–cell communication.

## 4. Discussion

Ovarian cancer is characterized by late-stage onset and a poor prognosis. Women frequently report “silent symptoms” such as stomach bloating and pain, resulting in a delayed referral for a malignancy workup [48]. Age at diagnosis, tumor stage, histological type, tumor grade, and the existence of residual illness after initial surgery are all known predictors of ovarian cancer. All of these known variables (except age) can only be evaluated after surgery [49,50]. As a result, the importance of developing early detection approaches and novel prognostic markers is being highlighted even more. Protein kinase C (PKC) is a prototypical class of serine/threonine kinases that signal via multiple pathways and regulate the expression of genes involved in cell cycle progression, tumorigenesis, and metastatic spread. Changes in the protein expression of PKC isozymes produce structural and functional changes that have been frequently connected to the presence of specific types of polymorphisms, resulting in the cancer-causing genetic sequence [51,52]. The purpose of this study is to determine the predictive impact of PRKCG polymorphism in patients with ovarian cancer.

Missense SNPs have been demonstrated to cause structural changes and boost oncogene function since they are situated in functional areas of genes. As a result, missense SNPs in specific genes may increase the likelihood of acquiring cancer. Single nucleotide polymorphisms (SNPs) in the PKC family have also been associated with several types of cancer [53,54]. Several studies have found a link between PKCγ and cancer at various stages, including glioblastoma, osteoblastoma and colon cancer, among others [55].

I-TASSER predicted the PRKCG protein model, which belongs to the conventional PKC (cPKC) class, which comprises PKCα, PKCβ1 and 2, and PKCγ. The model was chosen based on its C-score (−2.27) and InterPro prediction. In previous studies, I-TASSER was used to predict protein 3D models such as SARS-CoV-2, EZH2, TOX3, BRCA1, and many others [56,57,58]. I-TASSER was chosen based on an automated assessment of protein 3D structure prediction in CASP9, which takes a range of criteria into account to estimate predictor accuracy [59].

The connection between allele changes and ovarian cancer was explored after the rs923331350 and rs1331232028 Id SNPs were chosen in silico. To achieve this goal, two sets of primers (two outer and two inner) were constructed using Primer1 against the variants, and DNA was extracted. Following the extraction, tetra ARMS-PCR was performed. The statistical analysis using GraphPad revealed that when compared to homozygous GG and heterozygous AG, the results of PCR data showed that the rs1331232028 wild-type allele AA with the ODDs value of 2.50, relative risk of 1.53, and *p* value of 0.05 can be associated with ovarian cancer. The other alleles GG and AG have odds value and relative risk of less than 1, which makes them fall in the significant range. In contrast, the other SNP rs923331350 shows only one allele AG occurrence in both healthy and ovarian cancer patients, which makes the SNP null in wet lab analysis, and it cannot be associated with ovarian cancer. The wet lab analysis of PRKCG SNP rs1331232028, like many other SNPs in multiple genes, including ERCC1, XPC, PIK3CA, ERBB2 and ERCC2, can be linked to ovarian cancer susceptibility [60].

The effect of mutant SNPs of PKCγ (A24S and K359R) on the protein’s structure and function are checked by seeing the mRNA secondary structure stability. In the case of the A24S variant, the allele change does not change the mRNA stability, whereas in the K359R variant, the wild-type allele (A) causes less stability in the mRNA structure, which may cause a change in the protein’s structure and function. In addition, it can be associated with ovarian cancer progression, as genotype AA of variant K359R shows a link with ovarian cancer prognosis in wet lab analysis.

The variations in non-coding regions of the gene usually impact the expression of the gene at transcription as well as translation level [61]. Similarly, variations in the coding region influence the structure of the protein and ultimately its function [27]. Such structural modifications can also lead to the altered protein–protein interaction. PRKCE variations that are found to be associated with cancer stage and metastasis may also impact its interaction with other signaling proteins, leading to cancer progression.

Therefore, following an in vitro analysis of the SNP effect on the protein’s interaction capability, docking was performed utilizing HADDOCK. Connexin 43 was chosen to bind with PKCɣ because it directly phosphorylates Cx43 on Ser or Tyr residues, altering single-channel function and decreasing intercellular communication [62]. Cx43-mediated GJIC (Gap Junctional Intercellular Communication) that is overexpressed has been found to inhibit tumor growth by increasing cAMP transmission, implicating a role for Cx43–GJIC in carcinogenesis. Second, connexins may help to prevent cancer by interacting with signaling mediators via their C-terminal tails. As a result, Cx43 on the cell surface and Cx43 in the cytoplasm were both identified as tumor suppressors [63].

Among the docking clusters obtained, the cluster with a low Z-score was selected along with taking docking parameters into account, which include Van der Waals energy, electrostatic energy, desolvation energy, restraints violation energy and buried surface area. The docking results show more bonding of mutants as compared to the wild type, suggesting that mutant proteins interact more stably with connexin 43. Wild-type PKCγ down-regulates the gap junction by phosphorylating it, which causes selective permeability through them [64]. The docking analysis revealed the strong binding of mutant proteins with connexin 43, resulting in hyperphosphorylation of the gap junction which closes it, allowing no permeability at all.

After the docking result, MD simulations were conducted for the detailed analysis and further validation of SNPs on PKCγ–connexin 43 interactions. The RMSD value for wild-type interaction is high as compared to mutants, which shows the bonding is less stable as compared to mutants. Similarly, the gyration value of mutants is more than that of the wild type, indicating a strong interaction with connexin 43. The SASA analysis reveals more exposed surface area for wild-type interaction as compared to mutants, which means wild is having less interaction and thus more exposed areas. In contrast, the mutant having a strong association with connexin 43 has more buried surface area. The wild-type interaction has a loose interaction with connexin 43 as it has a larger gyration value, while both mutants show strong interaction with a smaller Rg value. The H-bonds remain almost the same in overall interaction. The bond length among the wild type throughout the simulation was also recorded, which changes, and the bond length decreased in the middle of the simulation. The bond length of mutants is less than of the wild-type interaction in the middle, which further usually reduces at the end of the simulation, showing the closer and more stable interaction of mutant PKCγ (A24S and K359R) with connexin 43. The docking analysis showing a more stable interaction of the mutant is validated further by simulation results. Thus, based on the MD simulation, we postulate that the mutation of residue A24S and K359R disrupts the coordination of gap junctions.

After docking and MD simulation along with the literature available through different databases, an interactive pathway is constructed to illustrate the effect of SNPs on cellular processes. In normal cells, gap junctions are present that allow the cell-to-cell communication, and PKCG is one of the down-regulators of the gap junction named connexin 43. PKC after activation through DAG and Ca^+2^ binds connexin 43 and phosphorylates it at Ser and Thr residues, making it semipermeable to small molecule passage [65]. The main connexins expressed in ovary cells are epithelium, Cx26, Cx43, and Cx32. Gap junctions are consistently down-regulated, absent, or not present in cell–cell communication. Ovarian carcinoma cells, like other cancers, have defects in intracellular and intercellular communication [43]. The SNPs in PKCγ A24S and K359R cause changes in binding with connexin 43 and may result in the closure of gap junctions by increasing the phosphorylation of residues, blocking the passage of small molecules and ions.

## 5. Conclusions

PKCγ disease-related variations were identified in silico, and likely pathogenic SNPs were chosen. PRKCG SNP rs923331350 showed no association with ovarian cancer progression as it shows null results in experimentation. While the A24S SNP shows null results in the Pakistani population, it could show an association with different populations around the world. Moreover, the sample data of our study belong to one particular area, and the sample size was small. The other chosen PKCγ protein missense mutation rs1331234028 variant allele AA was found to be harmful and highly related to ovarian cancer. The discovered SNP could be employed as a prognostic marker to aid in the early detection of ovarian cancer and act as a new potential therapeutic target. The expression profile of PKCγ following this mutation needs to be investigated, since it may open up new avenues in the cancer therapy field. Moreover, further research should be carried out for more validation of the results on a large sample size. The sample criteria could be expanded, and the marker can be checked with patients having any other cancer as well. The SNPs (A24S and K359R) of PKCγ can bind more strongly with connexin 43, which may lower its protein concentration in the cell. The effect of mutant PKCγ on the gap junction interactive pathway requires further validation, and more research should be carried out through in vitro and in vivo analysis.

## Figures and Tables

**Figure 1 genes-14-00236-f001:**
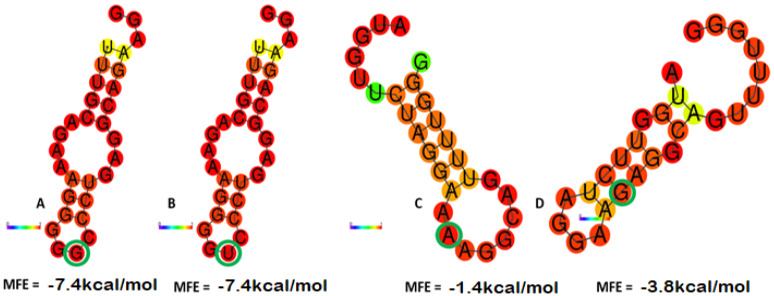
MFE value of mRNA secondary structures: (**A**) rs923331350 wild-type G allele; (**B**) rs923331350 variant U allele; (**C**) rs1331232028 wild-type A allele; (**D**) rs1331232028 variant G allele. Wild-type and altered alleles are highlighted with green circle.

**Figure 2 genes-14-00236-f002:**
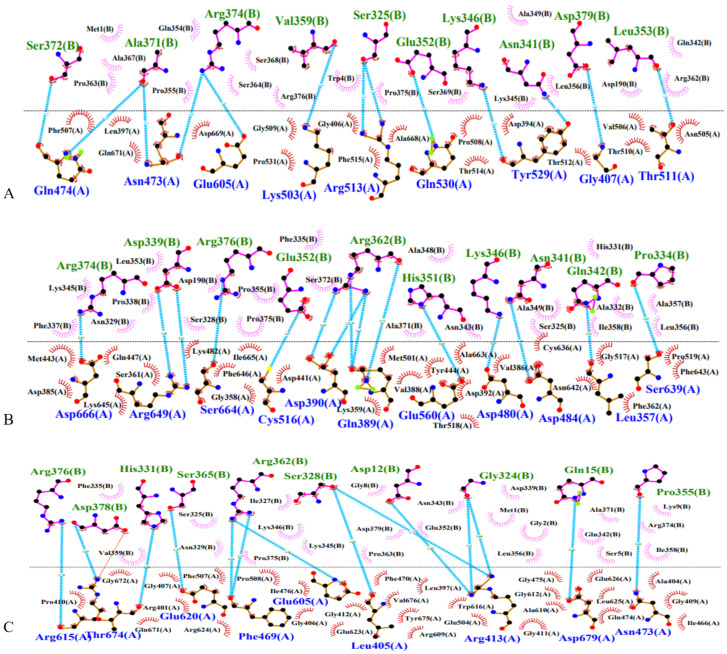
Post-docking interactions between active site residues of protein PKCγ with connexin 43: (**A**) Wild PKCγ interaction with connexin 43; (**B**) Mutant PKCγ (A24S) with connexin 43; and (**C**) Mutant PKCγ (K359R) with connexin 43.

**Figure 3 genes-14-00236-f003:**
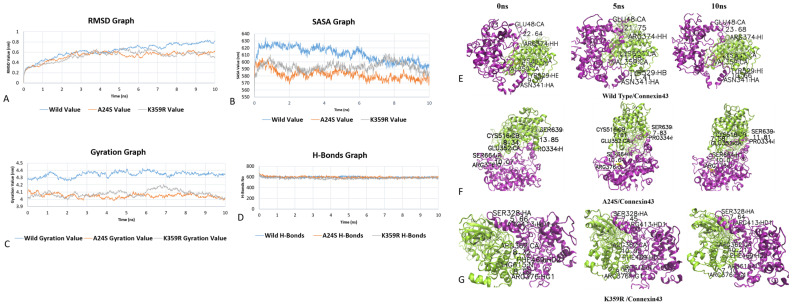
Analysis of simulation trajectory of native (Blue) and mutant structures A24S (Orange) and K359R (Gray) of PKCγ bound to connexin 43: (**A**) Root mean squared deviation (RMSD) plot for each trajectory over each 10 nanosecond production run, wild-type proteins have a higher RMSD value than mutant A24S and K359R. (**B**) The solvent-accessible surface area (SASA) analysis showing the more exposed surface area of wild and buried surface area for both mutants. (**C**) Radius of gyration of PKCγ and bound protein connexin 43 during the simulation run. Wild proteins have larger gyration values showing loose interaction, while both mutant A24S and K359R have smaller gyration values showing strong interaction with connexin 43. (**D**) H-bonds formed over the simulation run. (**E**) Two-dimensional (2D) view of the distance between hydrogen bond interaction of wild PKCγ (Green) with connexin 43 (purple) at different intervals. (**F**) Two-dimensional (2D) view of the distance between hydrogen bond interaction of Variant A24S PKCγ (green) with connexin 43 (purple) at different intervals. (**G**) Two-dimensional (2D) view of the distance between hydrogen bond interaction of variant K359R PKCγ (green) with connexin 43 (purple) at different intervals.

**Figure 4 genes-14-00236-f004:**
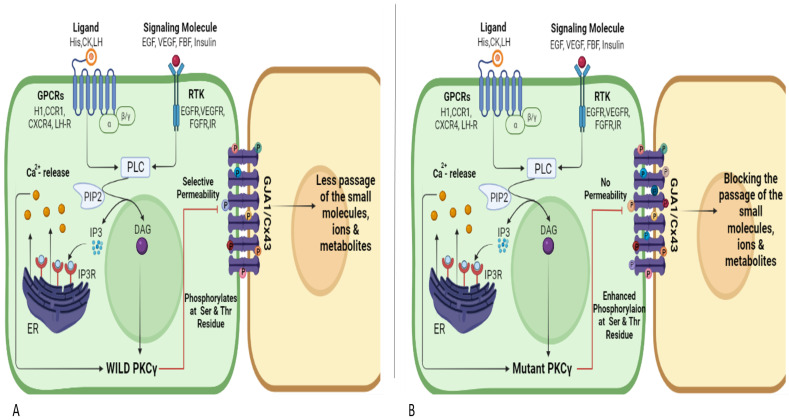
PKCγ regulating gap junctions: (**A**) Wild PKCγ causing semi-permeability through connexin 43 gap junction; (**B**) Mutant PKCγ completely closing the connexin 43 gap junction.

**Table 1 genes-14-00236-t001:** Sequences of primers and PCR conditions used for genotyping of PKCγ variants rs923331350 and rs1331232028.

Variant rs IDs	Primer Sequences	Denaturation	Annealing	Extension
rs923331350 G/T	Forward inner primer (G allele): GTTTTGCAGAAAGGAGG	95 °C	51 °C	72 °C
	Reverse inner primer (T allele): ACCTTCTGCCTCAGAGA			
	Forward outer primer (5′-3′): CTCGGAATTTCCCTGT			
	Reverse outer primer (5′-3′): AGTCGGGACTACAGCC			
rs1331232028 G/A	Forward inner primer (G allele): TTCCTCATGGTTCTAGGCAG	95 °C	57 °C	72 °C
	Reverse inner primer (A allele): ACCTTCCCAAAACTGCATT			
	Forward outer primer (5′-3′): GGTAGGAGGGTGGCCA			
	Reverse outer primer (5′-3′): CCGTCCCCTCAAGGAG			

**Table 2 genes-14-00236-t002:** Clinico-Pathological Characteristics of Patients.

Clinico-Pathological Characteristics of Patients		Ovarian Cancer (N) (%)
Age	≥50	23 (48)
	<50	26 (52)
Stage	I–II	17 (36)
	III-IV	32 (64)
Metastasis	Metastatic	19 (38)
	Non-metastatic	30 (62)
Treatment	Radiations	0 (0)
	Chemotherapy	49 (100)
	Radiations + Chemotherapy	1 (0)

**Table 3 genes-14-00236-t003:** Product Size for Genotype Bands.

Variants	Internal Band (Reference Allele)	Internal Band (Variant Allele)	Control Band
rs923331350 G/T	G-Allele	T-Allele	Outer
	224	197	387
rs1331232028 G/A	G-Allele	A-Allele	Outer
	224	291	476

**Table 4 genes-14-00236-t004:** Genotypic distribution of K359R (rs1331232028) SNP in ovarian cancer.

Genotype	Patient (*n* = 49)	Control (*n* = 51)	Odds Ratio	95% CI—Odds Ratio	Relative Risk	95% CI—Relative Risk	*p* Value
	(%)	(%)					
**AA**	20	11	2.508	1.059 to 5.989	1.535	1.027 to 2.230	0.0515
	40.82%	21.57%					
**AG**	15	14	1.166	0.4799 to 2.882	1.080	0.6801 to 1.609	0.8265
	30.61%	27.45%					
**GG**	14	26	0.3846	0.1631 to 0.9052	0.6000	0.3646 to 0.9337	0.0261
	28.57%	50.98%					
**A**	28	18	2.444	1.076 to 5.396	1.565	1.048 to 2.377	0.0442
	57.14%	35.29%					
**G**	21	33	0.4091	0.1853 to 0.9291	0.6389	0.4207 to 0.9538	0.0442
	42.86%	64.71%					

**Table 5 genes-14-00236-t005:** Association of K359R (rs1331232028) SNP with Metastatic State and Stage of Ovarian Cancer Patients.

Genotyping Distribution of rs1331232028 SNPs’ Clinical Features												
	Metastatic State						Cancer’s Stage					
Genotype	Metastatic			Non-Metastatic			Stage I-II			Stage III-IV		
	OR	RR	*p* Value	OR	RR	*p* Value	OR	RR	*p* Value	OR	RR	*p* Value
	(95% CI)	(95% CI)		(95% CI)	(95% CI)		(95% CI)	(95% CI)		(95% CI)	(95% CI)	
**AA**	0.41	0.52	**0.0017**	1.32	1.18	0.59	5.19	3.19	**0.0065**	1.58	1.30	0.44
	(0.13–1.20)	(0.22–1.14)		(0.43–3.55)	(0.60–2.09)		(1.53–16.89)	(1.42–7.12)		(0.61–4.08)	(0.71–2.17)	
**AG**	0.66	0.73	0.76	1.530	1.297	0.45	0.81	0.85	>0.99	1.38	1.21	0.62
	(0.21–2.28)	(0.27–1.72)		(0.55–4.04)	(0.71–2.23)		(0.25–2.99)	(0.31–2.07)		(0.51–3.57	(0.67–2.06)	
**GG**	6.23	3.503	0.12	0.5567	0.6885	0.25	0.20	0.28	**0.02**	0.43	0.59	0.11
	(2.01–19.43)	(1.62–7.63)		(0.21–1.46)	(0.37–1.22		(0.05–0.78)	(0.09–0.81)		(0.18–1.13	(0.31–1.05)	

**Table 6 genes-14-00236-t006:** Bond length among PKCγ variants and connexin 43.

Bond Length	Wild-Type			A24S Mutant			K359R Mutant		
	Lys503-Val359	Gln48-Arg374	Tyr529-Asn341	Ser639-Pro334	Ser664-Arg376	Cys516-Glu352	Arg413-Ser 328	Phe469-Arg362	Arg615-Arg376
0 ns	26.71 Å	15.67 Å	9.77 Å	13.85 Å	10.07 Å	8.34 Å	5.66 Å	8.80 Å	8.32 Å
5 ns	23.83 Å	11.08 Å	11.41 Å	7.83 Å	10.04 Å	7.61 Å	7.45 Å	8.59 Å	10.95 Å
10 ns	24.59 Å	17.07 Å	10.66 Å	11.81 Å	10.07 Å	7.58 Å	7.64 Å	7.10 Å	10.21 Å

## Data Availability

All the relevant data have been provided in the manuscript, and any Appendix A used and/or analyzed during the current study are available from the corresponding author on reasonable request.

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
