# Peer review of "Pathogenicity of PKCγ Genetic Variants—Possible Function as a Non-Invasive Diagnostic Biomarker in Ovarian Cancer"

_genes, 2023, doi:10.3390/genes14010236_

Round 1

Reviewer 1 Report

In this manuscript, Kanza et al. analyze the PKCγ genetic variant association with ovarian cancer and PKCγ’s protein interactions. The authors have identified the positive association of variant rs1331262028 and depicted the potential influence of variation on PKCγ molecular signaling with ovarian cancer. Here are my comments:

1. Many citations are reported with “Error! Reference source not found”: line 171, line 181, etc.

2. Line 233 Figure 1: please highlight G and U allele in Figure 1A and Figure 1B, respectively.

3. Line 136-line 140: Line spacing is different

4. The authors need to add a paragraph in Introduction part to explain why choose rs923331350 and rs1331232028 in PRKCG for analysis.

5. Line 384 Figure 4: the right borderline of the yellow box is covered

6. Please use ‘[]’ for the citation, not ‘()’. Because square bracket is more common

Author Response

Dear Editor

We are pleased to resubmit the revision of the article. We Have addressed all the comments raised by reviewers. The following are the answers

Reviewer 1

In this manuscript, Kanza et al. analyze the PKCγ genetic variant association with ovarian cancer and PKCγ’s protein interactions. The authors have identified the positive association of variant rs1331262028 and depicted the potential influence of variation on PKCγ molecular signaling with ovarian cancer. Here are my comments:

Response: Authors present their gratitude to the reviewer for the comments. We have addressed each comment below:

  1. Many citations are reported with “Error! Reference source not found”: line 171, line 181, etc.

Response: Thank you for pointing out the error. We have corrected the citation error in the sentences mentioned by the reviewer as well as in the whole manuscript (Lines 115, 175, 185, 198, 204, 228, 235, 250, 258, 265, 291, 312, 315, 336, 340, 344, 348, and 386)

  1. Line 233 Figure 1: please highlight G and U allele in Figure 1A and Figure 1B, respectively.

Response: In Figure 1, wildtype and altered alleles are highlighted in all parts of the figure.

  1. Line 136-line 140: Line spacing is different

Response: We have corrected the line spacing in mentioned paragraph and have ensured that the line spacing is uniform throughout the manuscript.

  1. The authors need to add a paragraph in Introduction part to explain why choose rs923331350 and rs1331232028 in PRKCG for analysis.

Response: We appreciate the reviewer’s positive comment. In our previous study, we used consensus tools to delineate the deleteriousness of PRKCG missense variants. Variants rs923331350 and rs1331232028 were indicated to be disease-causing and were also evaluated for their association with HCV-induced HCC (DOI doi.org/10.1186/s40364-022-00437-6). The present study is a continuation of that study and we have determined the association of both variants with ovarian cancer. The logic for opting for these variants for the study is described in lines 83-87.

  1. Line 384 Figure 4: the right borderline of the yellow box is covered

Response: Figure 4 depicts the influence of the PRKCG variants on the interaction between PRKCG and Connexin43 that ultimately affects the cell-to-cell connection. The yellow box in the figure is drawn partially so the connection between two cells can only be shown. Drawing a full yellow box might be confusing for the viewers. Furthermore, the space in a figure is also limited and by eliminating unwanted characters from the illustration we intended to present the actual idea more vividly.

  1. Please use ‘[]’ for the citation, not ‘()’. Because square bracket is more common

Response: We have revised the in-text citation of the whole manuscript and square brackets are used instead of round brackets.

Reviewer 2 Report

The authors analyzed the variant rs1331262028 in association with ovarian cancer and detected its impact on PKCγ’s protein interactions. There are few problems need to be solved by the authors:

  1. The authors are suggested to introduce some information about the role of rs1331262028 in cancer.
  2. The information of the 49 patients (such as whether the patients have other systemic diseases and whether they received chemotherapy etc.) need to be introduced in the section of “Materials and methods”.
  3. The underlying reason why rs1331232028 associated with metastatic state and stage of ovarian cancer need to be discussed in depth.

Author Response

Dear Editor

We are pleased to resubmit the revision of the article. We Have addressed all the comments raised by reviewers. The following are the answers

Reviewer 2

ï‚·  The authors are suggested to introduce some information about the role of rs1331262028 in cancer.

Response: In our previous research, we used consensus tools to assess the harmfulness of PRKCG missense variants. We found that variants rs923331350 and rs1331232028 were likely to cause disease and were also investigated for their relationship with HCV-induced HCC. In the current study, we are building on this previous work by examining the connection between these variants and ovarian cancer. The reasoning behind choosing these specific variants for the study is explained in lines 83-87 of the manuscript.

ï‚·  The information of the 49 patients (such as whether the patients have other systemic diseases and whether they received chemotherapy etc.) need to be introduced in the section of “Materials and methods”.

Response: Authors thank the reviewer for the constructive comment. Patients included in the study did not have any co-morbidity. Furthermore, all the patients were receiving chemo-/radiotherapy. This information is also described in the materials and methods section (lines 105-110).

ï‚·  The underlying reason why rs1331232028 associated with metastatic state and stage of ovarian cancer need to be discussed in depth.

Response: The reviewer’s comment allowed us to enhance the clarity of the manuscript. We have discussed the association of variant rs1331232028 with ovarian cancer stages and  metastatic state in the discussion section (lines 448-453).

Round 2

Reviewer 2 Report

In the revised article, the authors modified the manuscript referred to the comments, and answered the questions comprehensively.